# Nonlinear Robust Control of a Quadratic Boost Converter in a Wide Operation Range, Based on Extended Linearization Method

Francesco Alonge [1], Alessandro Busacca [1], Michele Calabretta [2], Filippo D'Ippolito [1], Adriano Fagiolini [1], Giovanni Garraffa [3], Angelo Alberto Messina [2,4], Antonino Sferlazza [1,*] and Salvatore Stivala [1]

1   Department of Engineering, University of Palermo, Viale delle Scienze Ed. 10, 90128 Palermo, Italy;
    francesco.alonge@unipa.it (F.A.); alessandro.busacca@unipa.it (A.B.); filippo.dippolito@unipa.it (F.D.);
    adriano.fagiolini@unipa.it (A.F.); salvatore.stivala@unipa.it (S.S.)
2   STMicroelectronics, Stradale Primosole, 50, 95125 Catania, Italy; michele.calabretta@st.com (M.C.);
    angelo.messina@st.com (A.A.M.)
3   Faculty of Engineering and Architecture, University of Enna KORE, 94100 Enna, Italy;
    giovanni.garraffa@unikore.it
4   Consiglio Nazionale delle Ricerche—Istituto per la Microelettronica e Microsistemi, Strada VIII,
    n. 5-Zona Industriale, 95121 Catania, Italy
*   Correspondence: antonino.sferlazza@unipa.it

**Abstract:** This paper proposes a control system for a quadratic boost DC/DC converter in a wide range of operations, based on an inner loop with a sliding mode controller, for reaching a desired equilibrium state, and an outer loop with integral-type controller, for assuring robustness against load and input voltage variations and converter parameter uncertainties. The sliding mode controller is designed with the extended linearization method and assures local asymptotic stability, whereas the integral controller is designed using classical frequency methods, and assures input–output stability. It is shown that the proposed controller also deals with the sudden changes in the nominal operating conditions; thus, if a change of the operating conditions takes place, the proposed control scheme automatically creates a sliding regime which stabilizes the converter trajectories to the new equilibrium point. Experimental results carried out on a suitably developed test set up show the effectiveness of the proposed approach.

**Keywords:** quadratic boost converter; sliding mode control; extended linearization; robustness

## 1. Introduction

Nowadays, the development of renewable power sources is driving interest towards DC–DC converters in particular for their control. Indeed, the majority of these sources are characterized by a low voltage output, i.e., photovoltaic panels, fuel cells, etc, as well as the majority of storage systems, i.e., batteries, supercapacitors, etc. This means that a DC–DC conversion stage is needed in order to allow the distribution of such energy [1]. For this reason, a relevant effort has been made to increase both the efficiency of DC–DC converters and their steady-state gain. The improvement of efficiency of this kind of system is important because it reduces the energy losses caused by switching and by the Joule effect. The steady-state gain is also relevant in terms of reducing the number of conversion stages when a high conversion ratio is required. A converter particularly suitable for the above-mentioned applications is the Quadratic Boost Converter (QBC), since it presents a high voltage ratio. Among the existing configurations of these converters, it is possible to distinguish two main topologies. The first one consists in two independent boost stages with two switching devices [2], while the second one is composed of a single conversion stage with only one switching device [3]. In this paper, attention is focused

on this last configuration, since the switching losses are reduced, allowing to potentially obtain a higher efficiency.

Among the control techniques proposed in the literature, sliding mode control is widespread, especially regarding single stage boost converters. For example, in [4] a sliding mode control law is designed for a boost converter according to a chosen sliding function. The implementation of this sliding mode controller leads to a converter control method with variable frequency. Since in practical applications it is requested that the converter is driven according to the PWM control method, the authors describe a conversion method from variable frequency control to a PWM control. The method is based on the interpretation of the equivalent control associated with the switching surface, as the duty cycle of the PWM control method. In [5], a robust sliding mode control law is proposed and experimentally tested for a boost converter. The robustness is achieved by means of an adaptive scheme which adjusts the parameters in order to cope with the variations of the load resistance and input voltage. The implementation of the controller is carried out as in [4], i.e., determining the equivalent control and using it as a duty cycle. In [6], a controller is proposed, consisting of an outer loop PI-type which gives the reference current, and a sliding mode inner control loop for the tracking of the reference current itself, using the tracking current error as sliding function. The implementation of the controller is carried out on a microprocessor-based device, and this allows the implementation of a predictive conversion strategy from the sliding mode control law to PWM control of the converter. In [7], a total sliding-mode Lyapunov-based control scheme is designed to ensure stable tracking performance under system uncertainties. There are also works that propose different interpretations of the sliding mode control for boost converters. For example, in [8] a sliding mode control technique is proposed for switched affine models, characterized by a continuous set of equilibrium states. The sliding function is obtained using an extended linearization method [9]. Simulation experiments show the peculiarities of the followed approach. In [10], a control technique is proposed for the same class of models, based on the minimization of an upper bound quadratic cost function involving the difference of the actual state and the desired equilibrium state. In [11], it is shown that the control strategy proposed in [8] is equivalent to a sliding mode strategy along a particular sliding surface, designed by means of a min-type algorithm.

With regard to quadratic boost converter, Refs. [12,13] propose a controller consisting of two control loops. In [12], the outer PI-type control loop regulates the output voltage, and gives the inductor reference current for the inner current loop. The inner control loop is designed using the low signal model corresponding to the averaged state space model. The control methodology used is the classical one in the frequency domain. In [13], a PI compensator regulates the output voltage giving the reference current for the inner loop. The inner loop is based on the sliding mode control of the current in the input inductor of the converter. The outer loop is designed using the classical approach in the frequency domain. In [14], a single quasi-resonant network that operates in a zero-current switching way is implemented with the aim of reducing the switching losses and obtain a higher conversion ratio. In [15], a hybrid control based only on the measurements of the input and output voltages is proposed by encompassing a control law and an observer for the estimation of the system states and, in particular, the inductor currents. Finally, in [16], starting from a hybrid model of the power converters, a PWM control algorithm is designed for the command of the converter, overcoming the well known approach based on the averaged state space model, having the duty cycle as input variable.

In this paper a sliding mode controller for a QBC is proposed but, differently from the papers [4–7], the sliding function is not based on the tracking errors of current or voltage, but it is more complex, depends on all the state variables, and is non-a priori defined, but it is derived from the design requirements and the employed control methods. Since the converter has to work in a wide range of equilibrium points, the sliding function is derived from the extended linearization approach [8,17]. Note that the sliding mode control of DC–DC power converters via extended linearization has been studied, from a theoretic

point of view, in few cases in the literature regarding a standard boost converter (cf., for example, ref. [17]) and a buck-boost power converter [18,19], but it has never been studied nor experimentally applied to a quadratic boost converter to the authors' best knowledge. Moreover, another design requirement is the robustness of the controller against parametric uncertainties and load and input voltage variations, and for this reason an outer loop with integral-type controller is designed.

The extended linearization method represents a systematic procedure of sliding mode controller design, where a nonlinear sliding surface, with well-defined properties, is designed on the basis of an extension of a linear sliding control carried out for affine linear models of converters. An important property of the proposed approach arises in the fact that if there is a sudden change in the nominal operating conditions, the control system automatically stabilizes, by means of a new sliding regime, the system trajectories of the new equilibrium point. The main advantage of this procedure is that the resulting sliding dynamics can be made linear by means of a suitable state coordinate transformation derivable from the linearized system model. The starting point is the construction of a model linearized around the desired equilibrium state and, assuming that it is controllable, the system is put in the control canonical form by means of a coordinate transformation. A linear sliding surface is chosen for this canonical form and then transferred into a nonlinear sliding surface in the starting coordinate frame. Finally, the integral controller of the outer loop updates the equilibrium state in order to maintain the output error at zero in the presence of load and input voltage variations, and converter parametric uncertainties. The implementation of the control law previously described is carried out with a constant sampling frequency. This frequency is chosen as the maximum allowable for computing the control law, and it depends on the computational power of the digital signal processor (DSP), and the complexity of the control law. Differently from almost all the papers cited before, the proposed control is not PWM-type. Indeed, the controller, at each sampling interval, decides if a commutation is required. Consequently, the switching frequency results vary and are less than (or equal to) the sampling frequency.

The paper is organized as follows. Firstly, a physical model is considered starting from the QBC circuit layout where the parasitic resistances of the inductances are considered. Then, a discontinuous fourth-order switching input-state-output mathematical model is provided. Given the discontinuous nature of the mathematical model, an affine LTI system is associated with it, which allows to determine the equilibrium states. Subsequently, the dynamics matrix of this model can be put in the companion form by means of an auxiliary input gain vector, and the corresponding coordinate transformation is computed. Then, a suitable sliding surface given by a linear combination of the state variables is defined, so that during the sliding motion on this surface the system is described by a linear, stable and autonomous third-order model. Lastly, the final sliding surface is expressed in terms of the current coordinates, obtaining a nonlinear sliding surface in terms of the coordinates of the original discontinuous model. The integral controller is designed using frequency domain control techniques so that the input–output stability of the whole control system in assured with a sufficient margin. The proposed control strategy has been tested experimentally in a suitable developer test set up, showing good closed loop behaviour.

## 2. Dynamic Model of the Quadratic Boost Converter

The electrical circuit of the QBC considered in this paper is shown in Figure 1. The converter can be modelled as a switching system consisting of a set of two models, defined by the status of the switch driving the converter itself, which is considered as an input and denoted by $u \in \{0, 1\}$. More precisely, $u = 0$ is associated with the status OFF, and $u = 1$ is associated with the status ON. Then, defining the state vector as follows:

$$\mathbf{x}^\top = \begin{bmatrix} i_{L1} & i_{L2} & v_{C1} & v_{C2} \end{bmatrix}, \tag{1}$$

the mathematical model of the QBC of Figure 1 is given by:

$$\dot{x} = A_{OFF}x + bV_{in} + A_1 xu, \tag{2}$$

$$y = c^\top x, \tag{3}$$

where $V_{in}$ is the input voltage, $y$ is the measured output and the matrices of the model are:

$$A_{OFF} = \begin{bmatrix} -\frac{r_{L1}}{L_1} & 0 & -\frac{1}{L_1} & 0 \\ 0 & -\frac{r_{L2}}{L_2} & \frac{1}{L_2} & -\frac{1}{L_2} \\ \frac{1}{C_1} & -\frac{1}{C_1} & 0 & 0 \\ 0 & \frac{1}{C_2} & 0 & -\frac{1}{C_2 R_0} \end{bmatrix}, \quad A_{ON} = \begin{bmatrix} -\frac{r_{L1}}{L_1} & 0 & 0 & 0 \\ 0 & -\frac{r_{L2}}{L_2} & \frac{1}{L_2} & 0 \\ 0 & -\frac{1}{C_1} & 0 & 0 \\ 0 & 0 & 0 & -\frac{1}{C_2 R_0} \end{bmatrix},$$

$$A_1 = A_{ON} - A_{OFF}, \quad b^\top = \begin{bmatrix} \frac{1}{L_1} & 0 & 0 & 0 \end{bmatrix}, \quad c^\top = \begin{bmatrix} 0 & 0 & 0 & 1 \end{bmatrix}.$$

Models (2) and (3) make up a switching model consisting of two linear time invariant (LTI) models described by the dynamic matrices $A_{OFF}$ when the status of the switch is OFF, and $A_{ON}$ when the status of the switch is ON. In both cases, the voltage $V_{in}$ is assumed to be constant. It is easy to verify that, since the inductors' parasitic resistances are taken into account, both models are asymptotically stable.

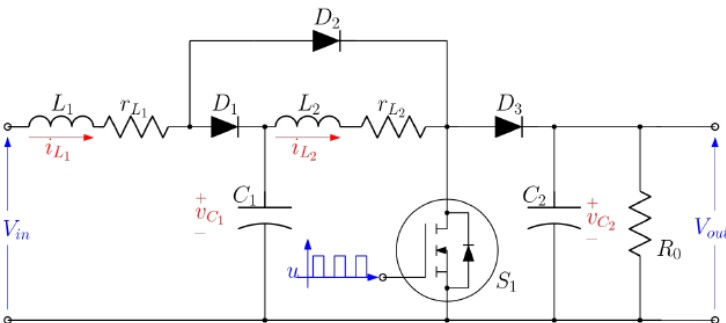

**Figure 1.** Electrical circuit of the quadratic boost converter.

In the contest of the switching models, it is important to define appropriately the equilibrium states. This can be made associating the switching model with an affine averaged model as illustrated in [8,17,20]. In particular, the affine averaged model, associated with (2) and (3), is given by:

$$\dot{z} = A(\lambda)z + bV_{in}, \tag{4}$$

$$y = c^\top z \tag{5}$$

where:

$$A = \lambda A_{ON} + (1 - \lambda) A_{OFF},$$

and lambda $\lambda \in [0,1]$. Note that matrix $A$ is a convex combination of the matrices $A_{ON}$ and $A_{OFF}$, and it is a Hurwitz matrix for $\lambda \in [0,1]$. Note that models (4) and (5) coincide with models (2) and (3) when $\lambda = u = 0$ (OFF state) and when $\lambda = u = 1$ (ON state).

Model (4) allows to determine the set of its equilibrium states $Z_e$. This set is defined as:

$$Z_e = \{z_e(\lambda) : Az_e + bV_{in} = 0 \wedge \lambda \in [0,1]\} \tag{6}$$

whose explicit expression is given by:

$$z_e(\lambda) = \frac{V_{in}}{g} \begin{bmatrix} 1 \\ (1-\lambda) \\ (1-\lambda)r_{L2} + (1-\lambda)^3 R_0 \\ (1-\lambda)^2 R_0, \end{bmatrix} \tag{7}$$

where $g = R_0(1-\lambda)^4 + r_{L2}(1-\lambda)^2 + r_{L1}$.

It is useful to note that the determinant of $A$, given by:

$$\det(A) = \frac{R_0(1-\lambda)^4 + r_{L2}(1-\lambda)^2 + r_{L1}}{C_1 C_2 L_1 L_2 R_0},\tag{8}$$

is always greater than zero for all $\lambda \in [0,1]$ and, consequently, there always exists an equilibrium point given by $z_e = -A^{-1}bV_{in}$. The output associated with $z_e$ is $y_e = c^\top z_e$.

Before the presentation of the control strategy, it is useful to study the structural properties of models (4) and (5). With regard to the observability properties, the system is not observable for all functioning modes; indeed, when the switch is in ON mode the observability matrix rank is equal to 1; therefore, the system is not observable in this mode. Nevertheless, the QBC is observable when it works in OFF mode because the observability matrix presents full rank for any value of the parameters. Note that this problem is the same one observed for the standard boost case [21] and it presents a physical interpretation. Indeed, it is easy to see in Figure 1 that, when the converter works in ON mode there is no information about the inductor currents $i_{L1}$ and $i_{L2}$ and about the voltage $v_{C1}$ that are being inferred from the output. The same considerations can be obtained by studying the reachability property; indeed, by computing the rank of the reachability matrix, it is not full-rank when the converter works in ON mode, while it is full-rank when it is in OFF mode. For all other values of $\lambda$ different from 0 and 1, $\lambda \in (0,1)$, the model (4) and (5) results are both observable and reachable.

## 3. Controller Design

The aim of this paper is to design a control system able to force the output of the QBC to reach a constant reference voltage, and to satisfy robustness properties despite load and input voltage variations. The control system consists of two control loops, an inner loop with a sliding mode controller which forces the state of the QBC model to reach the equilibrium state corresponding to the desired output voltage, and an outer loop controlled by a dynamical integral type controller which guarantees robustness against load and input voltage variations, and parametric uncertainties.

### 3.1. Inner Control Loop with Sliding Mode Controller

For the case under study, the sliding mode controller of models (2) and (3) is based on the assumption that the input voltage, load resistance and converter parameters assume the nominal values. It follows that $\lambda = \lambda_e$, where $\lambda_e$ is the value of the parameter $\lambda$ corresponding to the reference voltage, and the problem is the determination of a smooth function $s(x, z_e)$, and a control input $u$, such that:

$$u = \frac{1}{2}(1 + \text{sign}(s(x, z_e)),\tag{9}$$

where $\text{sign}(s(x, z_e)) = 1$ for $s(x, z_e) \geq 0$ and $\text{sign}(s(x, z_e)) = -1$ for $s(x, z_e) < 0$. The sliding surface $S_x$ is defined as $S_x = \{x \in \mathbb{R}^4 : s(x, z_e) = 0\}$. The function $s(x, z_e)$ has to be chosen such that $z_e \in S_x$ and the sliding condition:

$$s(x, z_e)\dot{s}(x, z_e) \leq -\eta|s(x, z_e)|\tag{10}$$

is satisfied, where $\eta$ is a positive constant. Indeed, if (10) holds, the state $x$ reaches the sliding surface $S_x$; it slides along it and reaches the desired equilibrium state $z_e \in S_x$. The sliding motion, ideally, occurs at an infinity frequency. During the sliding motion, Equation (2) does not represent the behaviour of the converter because the second member presents a non-numerable set of discontinuities in the input $u$ and, consequently, the Cauchy conditions for the existence and unicity of a solution are violated. To describe the dynamics of the system along the sliding modes, Filippov's method is used [22,23], which leads to

the affine averaged model (4) that, particularized for $s(x, z_e)$, can be used for describing the QBC during the sliding motion.

### 3.2. Dynamics of the Affine Model around the Desired Equilibrium State

Since there exist different equilibrium states, in order to determine the control strategy, it is convenient to express the affine dynamical model (4) by translating the origin of the state space into a generic equilibrium state $z_e^* = z_e(\lambda)|_{\lambda=\lambda_e}$. From (4) and (6), particularized for $\lambda = \lambda_e$, putting $\delta_z = z - z_e^*$ and $\delta_\lambda = \lambda - \lambda_e$, we obtain:

$$\dot{\delta}_z = A_e \delta_z + A_1 z_e^* \delta_\lambda + A_1 \delta_\lambda \delta_z, \tag{11}$$

where $A_e = \lambda_e A_{ON} + (1 - \lambda_e) A_{OFF}$. Model (11) is nonlinear, but for small neighbours of $z_e^*$ and $\lambda_e$, it can be linearized as follows:

$$\dot{\delta}_z = A_e \delta_z + A_1 z_e^* \delta_\lambda. \tag{12}$$

The linear model (12) is time-invariant, controllable, observable and asymptotically stable $\forall z_e^*$. Since, as already said, the design of the sliding mode controller is performed assuming $\lambda = \lambda_e, \delta_\lambda = 0$ and the model is also autonomous.

The problem now is to choose a sliding hyperplane $s(\delta_z) = \gamma^T \delta_z = 0$ for this model, which contains $z_e^*$. This hyperplane is designed as described in the following subsection. We just want to point out that the choice of this hyperplane corresponds to the choice of the dynamics of the closed loop system and it basically makes the difference with other sliding mode controllers proposed in the literature.

### 3.3. Structure of the Sliding Hyperplane for the Linearized Model (12)

In order to design the sliding hyperplane, the first step is to operate a coordinate transformation $\delta_z = T \hat{\delta}_z$ so that the dynamics matrix $\hat{A}_e = T^{-1} A_e T$ has the following canonical form:

$$\hat{A}_e = \begin{bmatrix} 0 & 1 & 0 & 0 \\ 0 & 0 & 1 & 0 \\ 0 & 0 & 0 & 1 \\ -a_0 & -a_1 & -a_2 & -a_3 \end{bmatrix}, \tag{13}$$

where the coefficients $a_i$, $i = 0, \ldots, 3$, are the coefficients of the characteristic polynomial of matrix $A_e$, given by:

$$a_0 = k_0 \frac{z_{e4}}{z_{e2}^2}, \quad a_1 = k_{01} + k_{02} \frac{z_{e4}}{z_{e3}}, \quad a_2 = k_{20} + k_{21} \frac{z_{e4}}{z_{e3}}, \quad a_3 = k_{30},$$

where:

$$k_0 = \frac{V_{in}}{(C_1 C_2 L_1 L_2 R_0) R_0}, \quad k_{01} = \frac{L_1 + R_0 r_{L1} C_2}{C_1 C_2 L_1 L_2 R_0}, \quad k_{02} = \frac{L_2 + C_1 R_0 r_{L1}}{(C_1 C_2 L_1 L_2 R_0) R_0},$$

$$k_{20} = \frac{C_2 L_1 R_0 + C_1 L_2 r_{L1}}{C_1 C_2 L_1 L_2 R_0}, \quad k_{21} = \frac{C_1 L_1 + C_2 L_2}{C_1 C_2 L_1 L_2 R_0}, \quad k_{30} = \frac{L_1 + C_2 R_0 r_{L1}}{C_2 L_1 R_0}.$$

In order to determine matrix $T$, it is convenient to consider a vector $h$ such that the couple $(A_e, h)$ is controllable and the structure of $h$ is as simple as possible. In the paper it is taken advantage from the controllability property of the couple $(A_e, b)$, and it is chosen as $h = b$. Then, matrix $T$ can be obtained from:

$$T^{-1} = \hat{Q}_c Q_c^{-1}, \tag{14}$$

where $Q_c$ and $\hat{Q}_c$ are the controllability matrices of the couples $(A_e, b)$ and $(\hat{A}_e, \hat{b})$, respectively, where $\hat{b} = \begin{bmatrix} 0 & 0 & 0 & 1 \end{bmatrix}^T$.

It is assumed that the sliding hyperplane is given by $\hat{s}(\hat{\delta}_z) = -\hat{c}^T \hat{\delta}_z$, and the sliding surface results:

$$\hat{s}(\hat{\delta}_z) = -\hat{c}^T \hat{\delta}_z = 0, \tag{15}$$

where $\hat{c}^T = \begin{bmatrix} c_1 & c_2 & c_3 & 1 \end{bmatrix}^T$. This implies that, during the sliding motion, the following relationship holds:

$$\hat{\delta}_{z4} = -c_1 \hat{\delta}_{z1} - c_2 \hat{\delta}_{z2} - c_3 \hat{\delta}_{z3}. \tag{16}$$

Consequently, the sliding regime is described by the third-order model given by:

$$\begin{bmatrix} \dot{\hat{\delta}}_{z1} \\ \dot{\hat{\delta}}_{z2} \\ \dot{\hat{\delta}}_{z3} \end{bmatrix} = \begin{bmatrix} 0 & 1 & 0 \\ 0 & 0 & 1 \\ -c_1 & -c_2 & -c_3 \end{bmatrix} \begin{bmatrix} \hat{\delta}_{z1} \\ \hat{\delta}_{z2} \\ \hat{\delta}_{z3} \end{bmatrix}, \tag{17}$$

which is asymptotically stable if the polynomial $\hat{\Delta}_s(p) = p^3 + c_3 p^2 + c_2 p + c_1$ is Hurwitz. In these conditions, $\hat{\delta}_z$ converges asymptotically to zero along $\hat{s}(\hat{\delta}_z) = 0$, whereas $\delta_z$ converges asymptotically to zero along $s(\delta_z) = \gamma^T \delta_z = 0$ with $\gamma^T = -\hat{c}^T T^{-1}$.

**Remark 1.** *The controllable canonical form model allows highlighting of the structure of the sliding hyperplane, the sliding surface* (15) *and the sliding regime* (17) *for the linearized model* (12) *with* $\delta_\lambda = 0$. *Moreover, it allows determining a simple stability condition that the coefficients of the sliding hyperplane have to satisfy for assuring the asymptotical stability of the sliding regime. Starting from the sliding hyperplane, the sliding function from the switching model* (2) *is obtained following the procedure illustrated in the next paragraph. From the behavioral point of view, whatever the desired equilibrium state is, the state of the system evolves around the above hyperplane after a transience of short duration, and at the steady state. This means that the sliding hyperplane is invariant with respect to the desired operating point.*

*3.4. Computation of the Sliding Function for the Switching Model* (2)

The problem now is that of determining a sliding function $s(x, z_e)$ for the switching model (2). According to the extended linearization method, this function has to contain the equilibrium state $z_e$, such that for any $\epsilon > 0$ there exists a $\delta > 0$ satisfying $\|x - z_e\| \leq \delta$, $\|s(x, z_e) - s(\hat{\delta}_z)\| \leq \epsilon, \forall x$. The last two requirements can be expressed as follows:

$$s(z_e, z_e) = 0, \tag{18}$$

$$\left. \frac{\partial s(x, z_e)}{\partial x} \right|_{x=z_e} = \gamma^T = -\hat{c}^T \frac{T_{adj}(z_e)}{\det(T)}, \tag{19}$$

where $T_{adj}$ is the adjoint matrix of $T$. Partitioning by columns matrix $T_{adj}$ is as follows:

$$T_{adj} = \begin{bmatrix} t_{adj,1} & | & t_{adj,2} & | & t_{adj,3} & | & t_{adj,4} \end{bmatrix},$$

Equation (19) can be split into four scalar equations:

$$\left. \frac{\partial s(x, z_e)}{\partial x_i} \right|_{z=z_e} = -\hat{c}^T \frac{t_{adj,i}(z_e)}{\det(T)}, \quad i = 1, \dots, 4. \tag{20}$$

Considering the model of the proposed converter and Equation (14), the elements $t_{adj,i}, i = 1, \dots, 4$, are:

$$t_{adj,1} = \begin{bmatrix} 0 \\ 0 \\ 0 \\ -k_{14T} \frac{z_{e4}}{z_{e2}^2} \end{bmatrix},$$

$$t_{adj,2} = \begin{bmatrix} k_{221Q}\frac{z_{e1}}{z_{e2}} \\ k_{231Q}\frac{z_{e1}}{z_{e2}} \\ 0 \\ \left(-k_{241T} - k_{242T}\frac{z_{e4}}{z_{e3}} - k_{243T}\frac{z_{e4}}{z_{e22}} - k_{244T}\frac{z_{e4}^2}{z_{e2}^2 z_{e3}}\right)\frac{z_{e1}}{z_{e2}} \end{bmatrix},$$

$$t_{adj,3} = \begin{bmatrix} \frac{1}{k_{231}}\frac{z_{e1}}{z_{e2}} \\ 0 \\ 0 \\ \left(-k_{341T} - k_{342T}\frac{z_{e4}}{z_{e3}} - k_{343T}\frac{z_{e4}}{z_{e22}}\right)\frac{z_{e1}}{z_{e2}} \end{bmatrix},$$

$$t_{adj,4} = \begin{bmatrix} k_{422Q} + k_{421Q}\frac{z_{e3}}{z_{e4}} \\ k_{431Q}\frac{z_{e3}}{z_{e4}} \\ \frac{1}{k_{442}}\frac{z_{e3}}{z_{e4}} \\ -k_{441T} - k_{442T}\frac{z_{e3}}{z_{e4}} - k_{443T}\frac{z_{e4}}{z_{e3}} - k_{444T}\frac{z_{e4}}{z_{e2}^2} - k_{445T}\frac{z_{e3}}{z_{e2}^2} \end{bmatrix},$$

where all the coefficients are defined in Appendix A.

Putting $\det(T(z_e)) = |\det(T(z_e))|\text{sign}(\det(T(z_e)))$, and multiplying both members of (19) by $\det(T(z_e))$, the following equation is obtained:

$$\left.\frac{\partial s^*(x, z_e)}{\partial x}\right|_{x=z_e} = -\text{sign}(\det(T(z_e)))\hat{c}^T T_{adj}(z_e), \tag{21}$$

where $s^*(x, z_e) = |\det(T(z_e))|s(x, z_e)$. Observing that $s^*$ and $s$ have the same manifold, i.e., $\{x : s^*(x, z_e) = 0\} = \{x : s(x, z_e) = 0\}$ and the same sign, it is convenient to choose $s^*$ as the sliding function. However, with a little abuse of notation, this function will be denoted again by $s$ in the following.

Equation (21) can be split into four equations:

That being stated, the sliding function can be obtained as follows:

$$s(z, z_e) = \sum_{i=1}^{4} \int_{z_{ei}}^{z_i} g_i(\xi_i)d\xi_i, \tag{22}$$

where

$$g_i = -\hat{c}^T \left.\frac{t_{adj,i}(z_e)}{\det(T)}\right|_{z_{ei}:=\xi_i}, \quad i = 1, \dots, 4,$$

$z_{ei}$ is the $i$-th component of the desired equilibrium state and it is a priori known and $z_i$ is the i-th component of the state vector of the converter model, and it is a measured quantity in the sampling instants. The explicit expressions of $g_i$, $i = 1, \dots, 4$, are given by:

$$g_1 = \frac{k_{14T}z_{e4}}{z_{e2}^2} \tag{23}$$

$$g_2 = \frac{z_{e1}(k_{244T}z_{e4}^2 + k_{243T}z_{e3}\xi_{e4})}{z_{e2}^3 z_{e3}} + \frac{k_{242T}z_{e1}z_{e4}}{z_{e2}z_{e3}} + \frac{z_{e1}(k_{241T} - k_{221Q}c_1 - k_{231Q}c_2)}{z_{e2}} \tag{24}$$

$$g_3 = -\frac{z_{e1}(c_1 z_{e3} + k_{231}k_{341T}z_{e3} - k_{231}k_{342T}z_{e4})}{k_{231}z_{e2}z_{e3}} - \frac{k_{343T}z_{e1}z_{e4}}{z_{e2}^3} \tag{25}$$

$$g_4 = k_{441T} - k_{422Q}c_1 + (k_{444T} + k_{445T})\left(\frac{z_{e3}}{z_{e2}^2}\right) + \left(k_{442T} - k_{421Q}c_1 - k_{431Q}c_2 + k_{443T} - \frac{c_3}{k_{442}}\right)\left(\frac{z_{e3}}{z_{e4}}\right) \tag{26}$$

and all the coefficients are given in Appendix A.

Finally, the candidate sliding function $s(x, z_e)$ for the discontinuous model describing the DC/DC quadratic boost converter is given by $s(x, z_e) = \sum_{i=1}^{4} s_i$, where:

$$s_1 = \frac{k_{14T}(x_1 - z_{e1})z_{e4}}{z_{e2}^2}, \tag{27}$$

$$s_2 = \log\left(\frac{x_2}{z_{e2}}\right)\left(k_{242T}\frac{z_{e4}}{z_{e3}} + \left(k_{241T} - k_{221Q}c_1 - k_{231Q}c_2\right)\right)z_{e1}$$
$$- \frac{k_{244T}z_{e1}z_{e4}^2 + k_{243T}z_{e1}z_{e3}z_{e4}}{2z_{e3}}\left(\frac{1}{x_2^2} - \frac{1}{z_{e2}^2}\right), \tag{28}$$

$$s_3 = z_{e1}\left(\left(k_{343T}z_{e4} + k_{341T}z_{e2}^2\right)(x_3 - z_{e3}) + k_{342T}z_{e2}^2 z_{e4}\log\left(\frac{x_3}{z_{e3}}\right)\right)\frac{1}{z_{e2}^3} - \frac{c_1 z_{e1}(x_3 - z_{e3})}{k_{231}z_{e2}}, \tag{29}$$

$$s_4 = \left(k_{441T} - k_{422Q}c_1 + \frac{k_{445T}z_{e3}}{z_{e2}^2}\right)(x_4 - z_{e4})$$
$$\left(k_{442T}z_{e3} - k_{421Q}c_1 z_{e3} - k_{431Q}c_2 z_{e3} - \frac{c_3 z_{e3}}{k_{442}}\right)\log\left(\frac{x_4}{z_{e4}}\right)\left(\frac{k_{443T}}{2z_{e3}} + \frac{k_{444T}}{2z_{e2}^2}\right)(x_4^2 - z_{e4}^2). \tag{30}$$

To ensure the existence of a sliding motion in a neighbourhood of the surface $s(x, z_e) = 0$, it should be ensured that the derivative of the sliding function $s(x, z_e)$ with respect to $x$ points towards the sliding surface. Mathematically, this implies that condition (10) is satisfied. Condition (10) can be written as $\dot{s}(x, z_e)\text{sign}(s(x, z_e)) \leq -\eta$, where:

$$\dot{s}(x, z_e) = \frac{\partial s}{\partial x}\left(A_{OFF}x + bV_{in} + \frac{A_1}{2}x(1 + \text{sign}(s(\cdot)))\right). \tag{31}$$

From (31), the following conditions are obtained:

$$\frac{\partial s}{\partial x}\left(A_{ON}x + bV_{in}\right) \leq -\eta, \quad \text{if } s > 0, \tag{32}$$

$$\frac{\partial s}{\partial x}\left(A_{OFF}x + bV_{in}\right) \geq \eta, \quad \text{if } s < 0. \tag{33}$$

Alternatively, the sliding mode existence conditions can be verified using the equivalent control method. As is known, the equivalent control, $u_{eq}$, forces the state $x$ to slide on this sliding surface. This implies that $\dot{s}(x, z_e^*) = 0$, and consequently, the following equation holds along the state trajectories:

$$\dot{s}(x, z_e^*) = \nabla(s)(A_{OFF}x + bV_{in} + A_1 x u_{eq}) = 0, \tag{34}$$

where $\nabla(s) = \frac{\partial s}{\partial x}$. It follows that the equivalent control is:

$$u_{eq} = -\frac{\nabla(s)(A_{OFF}x + bV_{in})}{\nabla(s)(A_1 x)}. \tag{35}$$

From [24] (Theorem 2.9), it can be concluded that necessary and sufficient condition for the local existence of a sliding regime over the manifold $S$ is:

$$0 < u_{eq} < 1. \tag{36}$$

This means that around an arbitrarily small neighbourhood of the sliding surface, a sliding motion takes place, and due to the construction of the sliding function $s(x, z_e^*)$ (cf. (18) and (19)), the state converges to the equilibrium state very quickly. In fact, the result-

ing sliding function is a nonlinear function of the state variables, the desired equilibrium state and the converter parameters, and it does not contain dynamical terms.

In this regard, it is interesting to compute numerically the equivalent control for $x = z_e^*$ as a function of the converter output voltage, and to evaluate whether condition (36) is satisfied. For example, for the QBC whose parameters are shown in Table 1, the characteristic equivalent control vs. output voltage is illustrated in Figure 2. Examination of this figure shows that the equivalent control computed in the equilibrium state is sufficiently far from the bounds 0 and 1 when the output voltage ranges from 30 to 500 V. This was further confirmed in various simulation experiments.

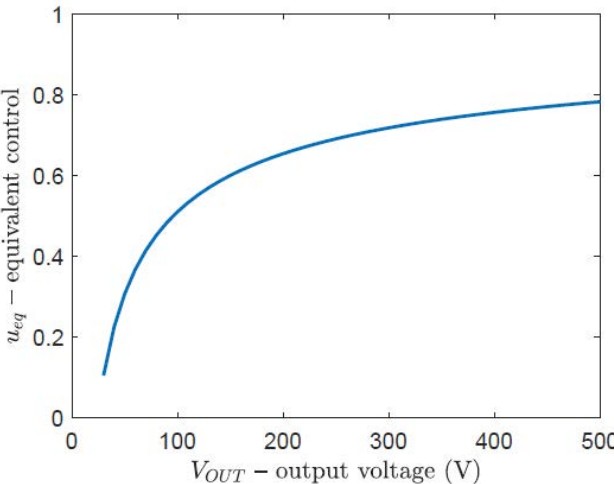

**Figure 2.** Equivalent control vs. output voltage.

**Table 1.** Parameters of the QBC.

| Component | Value | Model | Description |
|---|---|---|---|
| $V_{IN}$ | 24 V | | Input Voltage |
| $L_1$ | 330 μH | AGP4233-334ME | Inductor |
| $L_2$ | 470 μH | AGP4233-474ME | Inductor |
| $r_{L_1}, r_{L_1}$ | 11.5 mΩ | | Equivalent series resistance of the inductors |
| $C_1, C_2$ | 20 μF | MKP1848C62090JP4 | Capacitors |
| $r_{C_1}, r_{C_1}$ | 5 mΩ | | Equivalent series resistance of the capacitor |
| $R_0$ | 380 Ω | | Load Resistor |
| $D_{1,2,3}$ | | C3D06060A | Diodes |
| $S_1$ | | C3M0065090D | Switch |
| Driver | | 1EDI20N12AF | Switch Driver |

**Remark 2.** *Note that the sliding surface presented in this paper is suitably developed for the considered QBC and is not presented in other works in the literature to the best of the authors' knowledge. Indeed, the extended linearization procedure was only applied to simpler converter topologies, such as the boost converter, which results in easier sliding surfaces, and is tested only in simulation, whereas, in this work, experimental results will be shown. Finally, the above procedure for determining the matrix $T$, which allows to extend the sliding hyperplane designed in a neighborhood of the $z_e^*$ to the sliding function for the nonlinear model, is a further very important contribution of the paper.*

### 3.5. Controller Scheme and Design of the Outer Control Loop

The inner loop block scheme, consisting of the sliding mode controller and the DC/DC converter is shown in Figure 3. The reference voltage is the input of the system and the value of $\lambda_e$, useful to obtain $z_e$, is computed by the following relation:

$$\lambda_e = 1 - \sqrt{\frac{-h + \sqrt{h^2 - 4V_{out}^2 r_{L1} R_0}}{2V_{out} R_0}}, \tag{37}$$

where $h = r_{L2} V_{out} - R_0 V_{in}$, and $V_{out}$ represents the reference output voltage. The equilibrium state, obtained by (7) with $\lambda = \lambda_e$ ($\lambda_e$ computed by (37)) represents the input of the control loop and the feedback signal is the state $x$ of the QBC. The sliding mode controller establishes the input $u \in \{0, 1\}$, according to (9). Note that the value of $\lambda_e$ is constant once the equilibrium point is chosen, and it changes only if the reference value of the output voltage has to be changed.

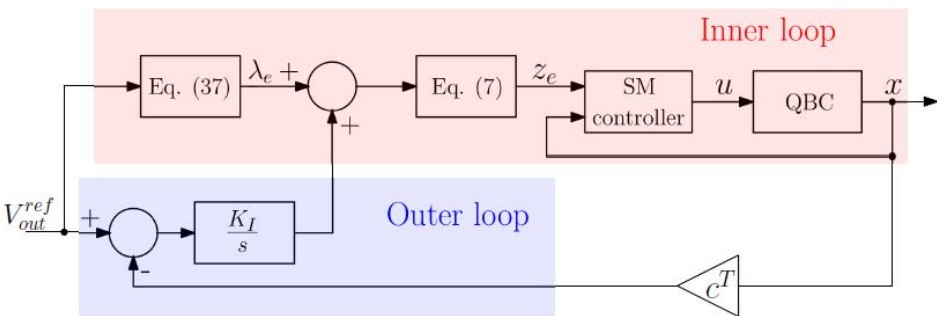

**Figure 3.** Block diagram of the control system.

As already said, the task of the outer control loop is that of giving robustness to the whole closed loop system. Looking at (7), it appears that the equilibrium state for a given value of $\lambda$ depends on the load, input voltage and converter parameters. Consequently, if one or more of these quantities vary with respect to their nominal values, the equilibrium state varies with respect to that desired. An efficient way to cope with these variations is that of constructing an outer control loop, driven by the difference between the reference output voltage and the measured one, able to give the value of $\Delta_\lambda$, to add to $\lambda_e$, so that the output voltage converges to the reference one.

The outer loop controller is integral-type ($K_I/s$) and the block scheme of the whole control system is given in Figure 3. The gain $K_I$ is obtained as follows. Assuming $\Delta_\lambda \ll \lambda_e$ and $\|\delta_z\| \ll 1$, the linearized model of the plant consists of (12), and the output equation is $\delta_y = y - y_e = c^T \delta_z$, where $y_e = c^T z_e^*$. Assuming as plant the transfer function from $\delta_\lambda$ to $\delta_y$, the gain $K_I$ can be obtained using frequency domain control design techniques in order to ensure asymptotic stability with a sufficient margin and a sufficient value of crossover frequency. However, since the transfer function varies with the desired equilibrium state, it is necessary to adjust the gain of the integral action in order to obtain the desired stability margins and crossover frequency. In practice, it is convenient to construct, off-line, a lookup table that contains the couples $K_I, V_{out}^{ref}$. Considering the QBC whose parameters are shown in Table 1, Figure 4 contains the waveform of the integral gain value as function of the output voltage that allows to obtain a crossover frequency of 100 rad/s, a gain margin of 8.4 dB and the phase margin is $m_\phi = 88.7$ degrees in the range 40–500 V. This waveform can be used to update online the integral gain such that the desired stability margins and crossover frequency are satisfied in a wide range of output voltage variations.

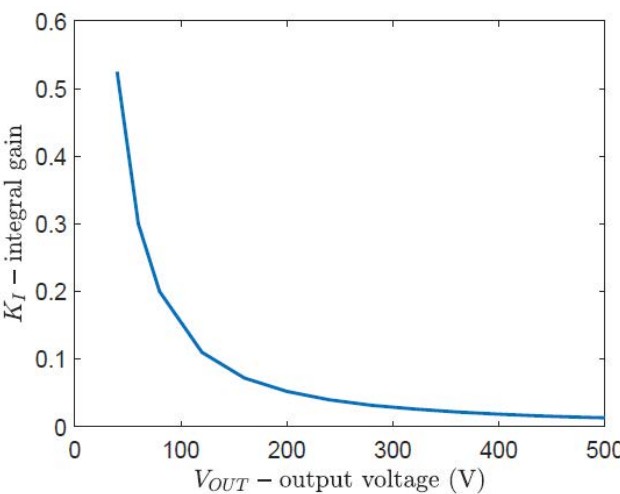

**Figure 4.** Integral gain value vs. output voltage.

## 4. Experimental Set-Up

A test setup has been suitably built to validate the proposed control technique. The general architecture of the experimental set-up follows the schemes shown in Figure 3.

The converter under test is shown in Figure 1 and the parameters, as well as the used components, are given in Table 1. A photo of the test bench is shown in Figure 5.

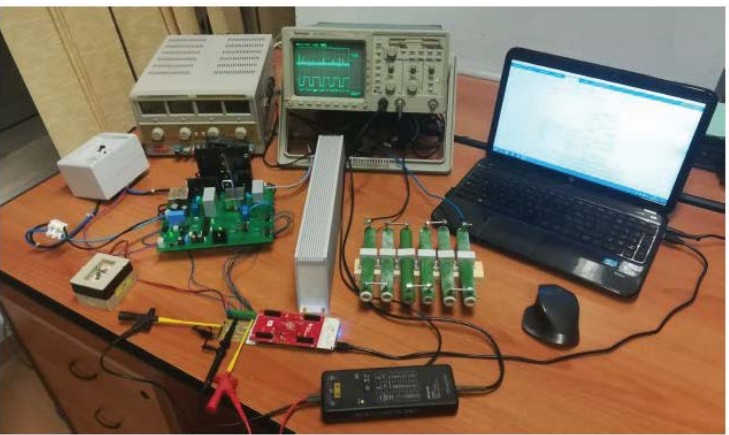

**Figure 5.** Photo of the experimental test bench.

The controller was digitally implemented using the C2000 32-bit TI microcontroller TMS320F28379D that has an additional built-in dedicated processor acting as the control law accelerator (CLA). In particular, the implementation is developed so that the SM algorithm runs in the dedicated CLA CPU while the main CPU takes account of other non-real-time tasks. Due to the variable frequency nature of the control law, the digital controller was implemented by directly driving the DSP digital output connected to the switch instead of using a PWM module that runs at a fixed frequency. To accomplish this task, the peculiar architecture of the TMS320F28379D was exploited. In particular, the control law was executed on the CLA (control law Accelerator), which takes 3.125 μs to execute the ADC conversion and to evaluate whether a change in the digital output must be imposed. This time can be interpreted as a dwell time for the commutation. With this implementation strategy, the maximum achievable frequency results in 320 kHz. In any case, this does not mean that the switching frequency is always the maximum; indeed, if the SM control strategy does not need to impose a change in the state of the switch, no transactions are imposed on the DSP digital pin, thus resulting in a switching frequency lower than the maximum one of 320 kHz.

The inductor currents $I_{L1}$ and $I_{L2}$ are measured by means of Hall-effect sensors LEM LTS-15-NP, while the $V_{C1}$ and $V_{OUT}$ voltages are measured by means of a voltage divider and an operational amplifier LM324 in buffer configuration. All signals are sampled and converted by the four embedded analogues to digital converters and processed by the microcontroller.

## 5. Experimental Results

In this section, experimental results are given. In particular, a start-up test, a load variation test and a supply voltage variation test were carried out to validate the effectiveness of the proposed controller.

Figures 6–8 are relative to the behavior of the system with the proposed control scheme of Figure 3 without the outer loop. In particular, Figure 6 shows $V_{C1}$ and $V_{out}$ voltages, Figure 7 shows the waveforms of the inductor currents $I_{L1}$ and $I_{L2}$, while Figure 8 shows the behaviour of the converter at the steady state. From these figures, it is evident that the converter behaves very well; indeed, the current and voltage waveforms reach their steady-state values very fast (about 2 ms), with almost null voltage overshoot and a very limited inrush current. These current overshoots are due to the fact that the output capacitor is discharged at the beginning of the experiments. For this reason, in order to obtain a fast output voltage response a high current is required. In any case, the current is limited and well tolerated by the converter. This confirms the expected results. In particular, the proposed algorithm allows one to control all the state variables (instead of the output variable only), exhibiting an instantaneous control action. Moreover, from Figure 8, it is possible to appreciate a good voltage regulation at a steady state, with small ripple (about $\pm 1.5$ V), and the current waveform shows the variable frequency behaviour of the proposed control strategy. Nevertheless, the current excursion is limited and it exhibits a "mean switching frequency" of 60 kHz. In order to evaluate the steady-state performance, Figure 9 depicts the measured system efficiency for different values of input power. A good efficiency can be observed, about 95%, under nominal operating conditions.

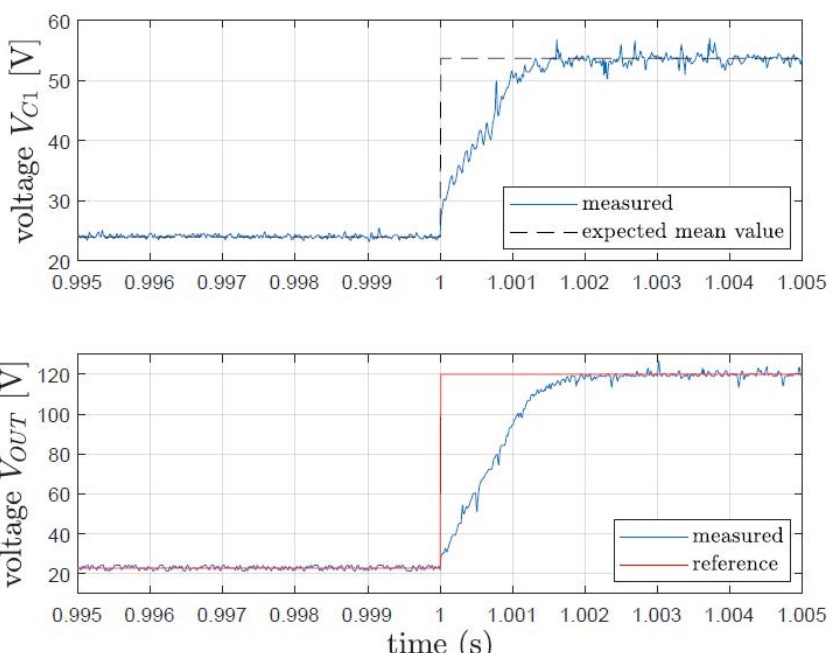

**Figure 6.** $V_{C1}$ and $V_{out}$ voltages during a start-up test, without integral action.

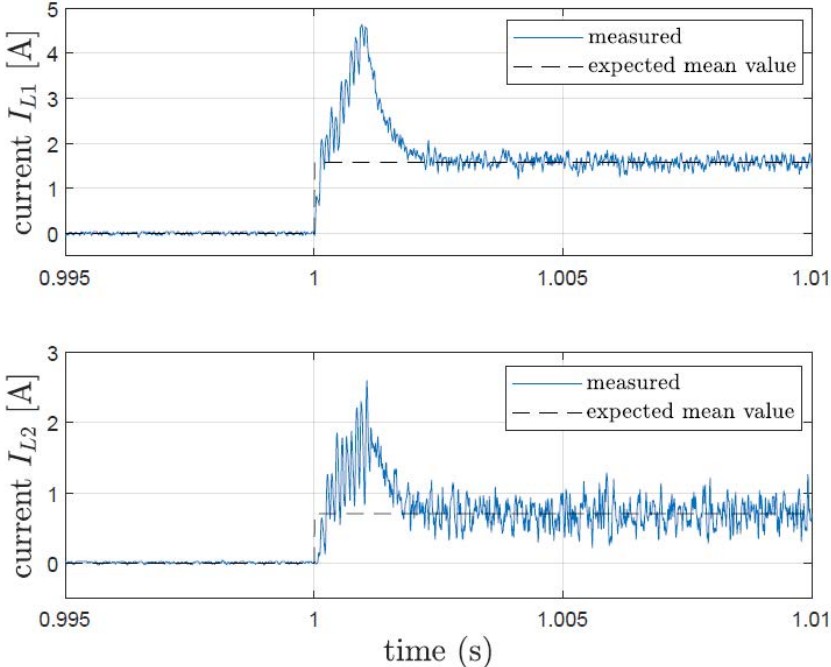

**Figure 7.** $I_{L1}$ and $I_{L2}$ currents during a start-up test, without integral action.

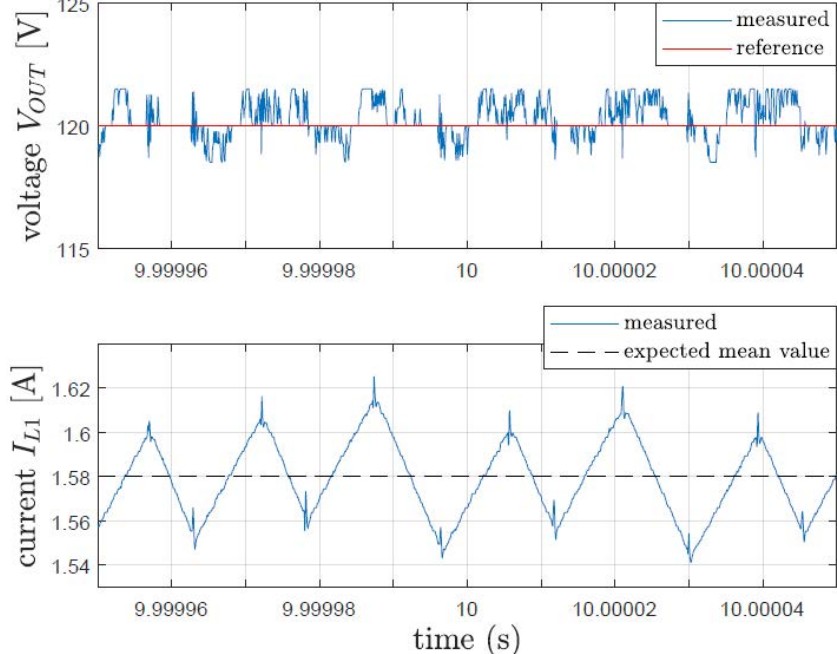

**Figure 8.** $I_{L1}$ and $I_{L2}$ currents at steady state.

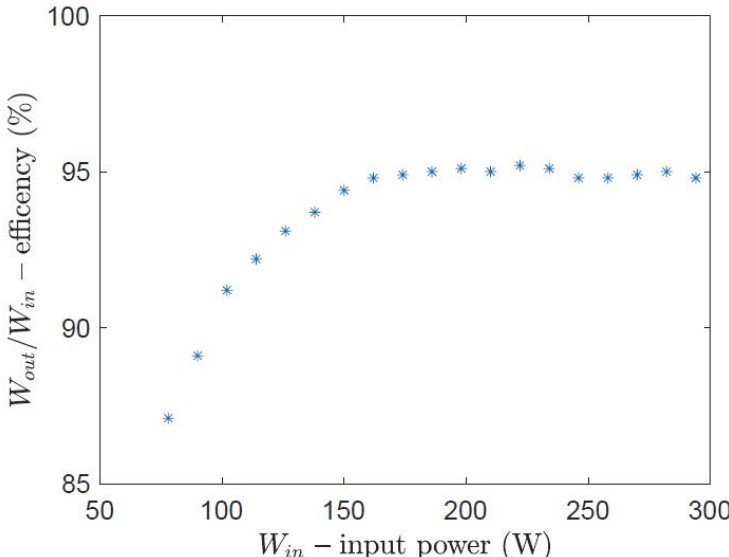

**Figure 9.** Efficiency of the system at steady state for different values of input power.

In order to verify the effect of the outer loop, with integral action, the same start-up test was repeated using the control scheme of Figure 3 with outer loop, and the results are given in Figures 10 and 11. By comparing Figure 6 with Figure 10 and Figure 7 with Figure 11, it is evident that the external loop does not affect the system performance in terms of settling time and dynamic precision. Indeed, when we are working under nominal conditions, the external loop is as if it is disabled and the dynamics is imposed by the internal sliding mode loop. The external loop intervenes only when an unknown variation of load, input voltage or parameters takes place. In this last case, i.e., if unknown load and/or supply voltage variations or any other parameter variation occurs, an undesirable steady-state error takes place. For this reason, the outer loop, where the integral action is added in order to deal with this drawback, gives robustness to the proposed strategy. By means of this control scheme, load and supply voltage variation tests were carried out.

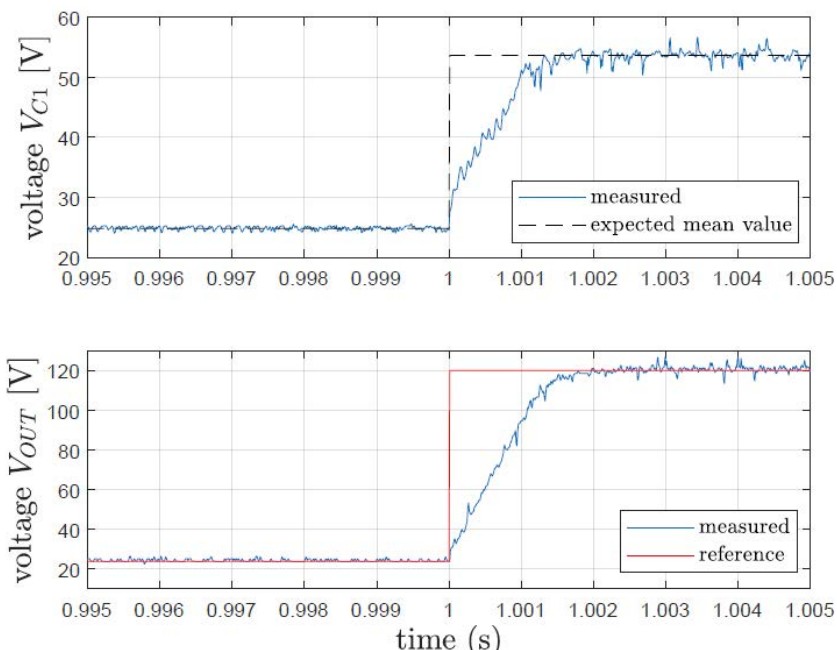

**Figure 10.** $V_{C1}$ and $V_{out}$ voltages during a start-up test, with integral action.

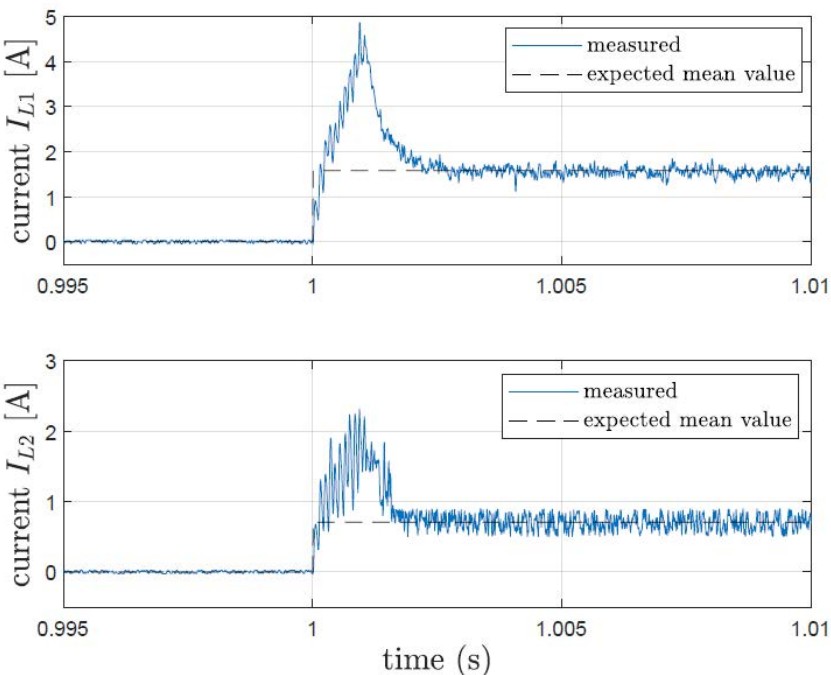

**Figure 11.** $I_{L1}$ and $I_{L2}$ currents during a start-up test, with integral action.

In particular, with regard to the load variations, Figure 12 shows the voltages $V_{C1}$ and $V_{out}$ and Figure 13 shows the waveforms of the inductor currents $I_{L1}$ and $I_{L2}$ during a load variation from $R_0 = 380 \ \Omega$ to $R_0 = 220 \ \Omega$. From these figures, the fast response of the system is evident. Moreover, due to the presence of the integral action, the steady-state error is totally compensated for. The only drawback coming from the use of the integral action is an increment of the settling time which is about 20 ms in this test, results higher than the one in the start-up test (2 ms). This is because the inner loop is faster than the outer loop.

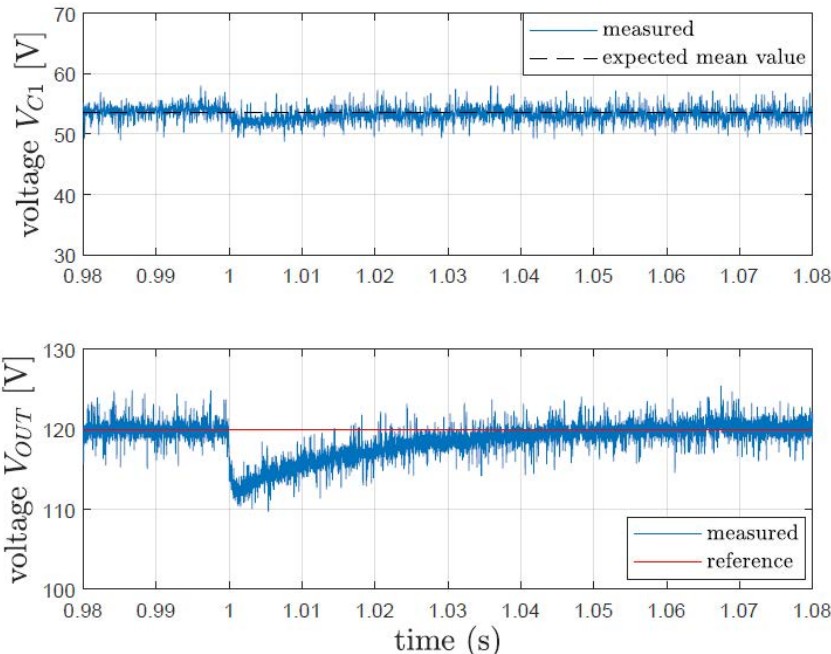

**Figure 12.** $V_{C1}$ and $V_{out}$ voltages, during a load variation from $R_0 = 380 \ \Omega$ to $R_0 = 220 \ \Omega$, with integral action.

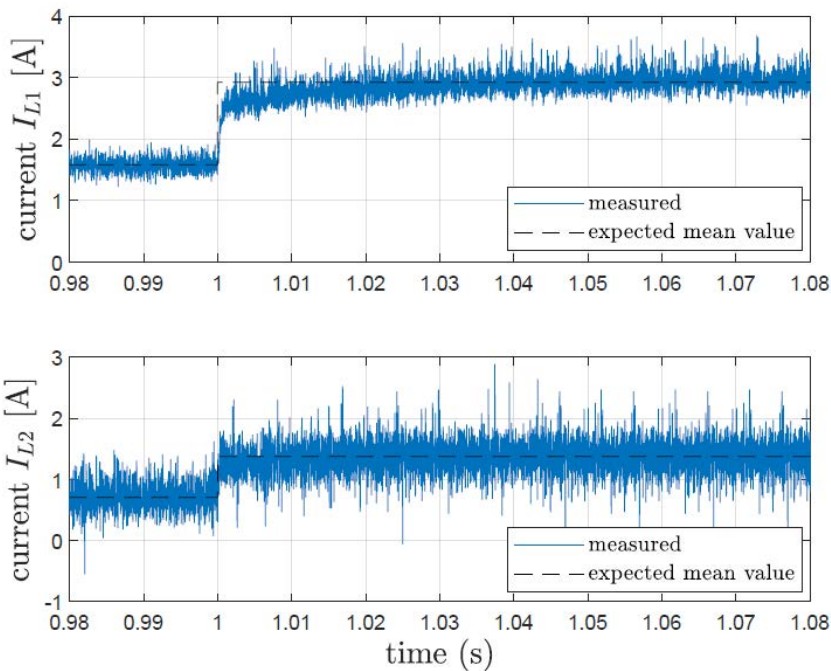

**Figure 13.** $I_{L1}$ and $I_{L2}$ currents, during a load variation from $R_0 = 380\ \Omega$ to $R_0 = 220\ \Omega$, with integral action.

With regards to the supply voltage variations, Figure 14 shows voltages $V_{C1}$ and $V_{out}$ and Figure 15 shows the waveforms of the inductor currents $I_{L1}$ and $I_{L2}$ during a supply voltage variation from $V_{IN} = 24$ V to $V_{IN} = 20$ V. Even in this case, the experimental results show that the system is able to deal with sudden (and unknown) supply voltage variations simulating, for example, a battery pack cell fault or shadow operating condition on a solar panel. In this test the resulting settling time is about 15 ms with an undershoot on the output voltage less than 5%, which is negligible.

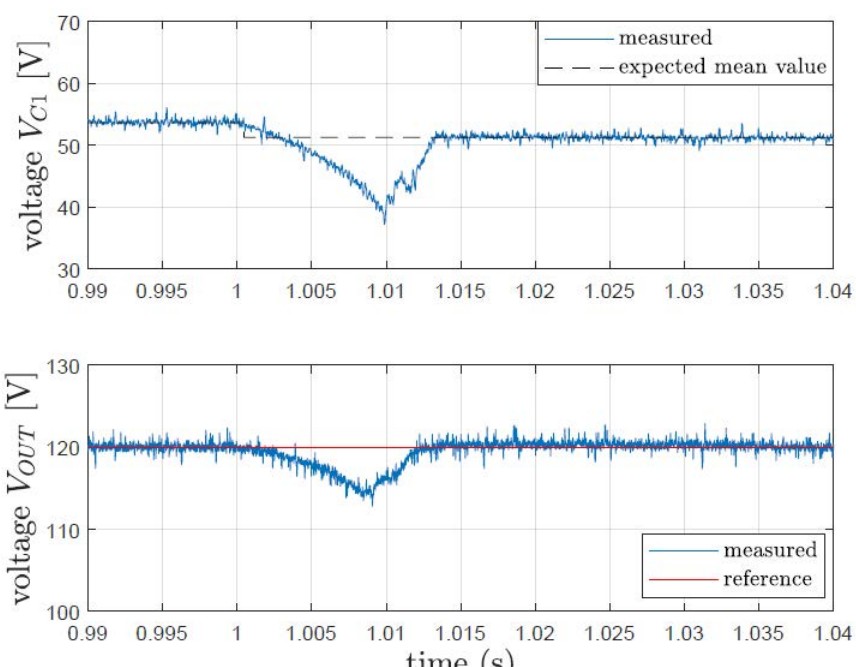

**Figure 14.** $V_{C1}$ and $V_{out}$ voltages, during a supply voltage variation from $V_{IN} = 24$ V to $V_{IN} = 20$ V, with integral action.

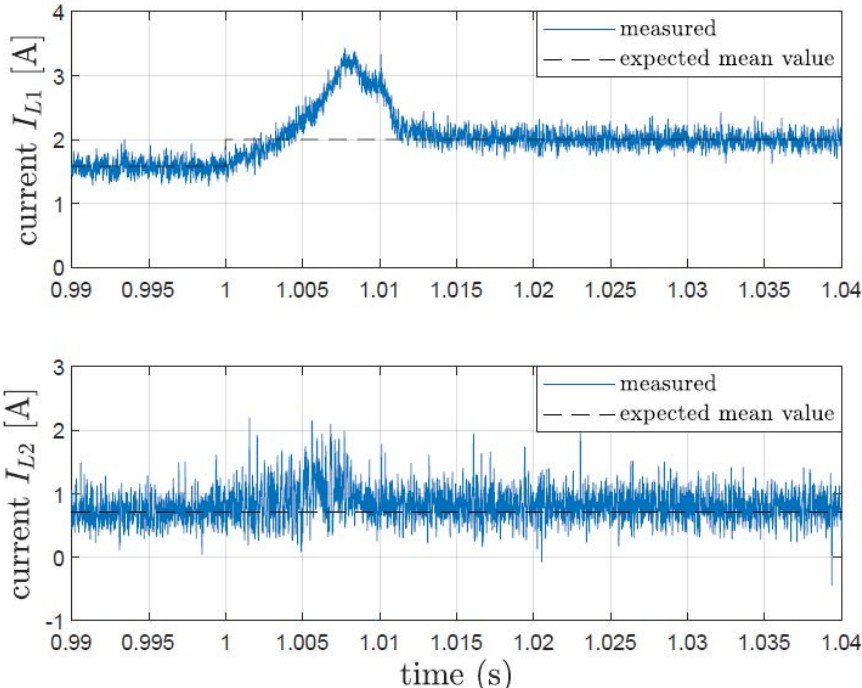

**Figure 15.** $I_{L1}$ and $I_{L2}$ currents, during a supply voltage variation from $V_{IN} = 24$ V to $V_{IN} = 20$ V, with integral action.

All these results are summarized in Table 2, where the main performance indexes (settling time and overshoot) are shown for all tests.

**Table 2.** Performance indexes.

|  | Settling Time | | | | Overshoot | | | |
|---|---|---|---|---|---|---|---|---|
| **Test** | **Current** | | **Voltage** | | **Current** | | **Voltage** | |
|  | $I_{L1}$ | $I_{L2}$ | $V_{C1}$ | $V_{OUT}$ | $I_{L1}$ | $I_{L2}$ | $V_{C1}$ | $V_{OUT}$ |
| Start up | 2.5 ms | 2.1 ms | 2 ms | 2.2 ms | 3 A | 1.6 A | 0 V | 0 V |
| Load Variation | 30 ms | 5 ms | 25 ms | 40 ms | 0 A | 0 A | 3.5 V | 9 V |
| input Variation | 14 ms | 11 ms | 14 ms | 13 ms | 1.3 A | 0.5 A | 12 V | 6 V |
| Start-up with PI | 4.3 ms | 3.9 ms | 3.8 ms | 4 ms | 18 A | 3.6 A | 4.5 V | 9 V |

Finally, in order to compare the proposed technique with a conventional control strategy, a standard PID controller with constant PWM was implemented, and the same start-up test was carried out. In particular, the PID controller was tuned in order to obtain almost the same settling time obtained with the proposed strategy. Figure 16 shows the output voltage $V_{out}$, while Figure 17 shows the $I_{L1}$ and $I_{L2}$ currents during the same test. From these figures, the differences with the previous control strategy are evident. Indeed, the currents present a high overshoot that is not present in the currents of Figures 7 and 11. Moreover, the output voltage presents a worst transience with respect to the one of Figures 6 and 10, because from the comparison, a voltage overshoot is evident in Figure 16 that is not present in Figures 6 and 10, and even the settling time results are higher. All these results are summarized in Table 2, where the numerical values of the overshoot and of the settling time are shown.

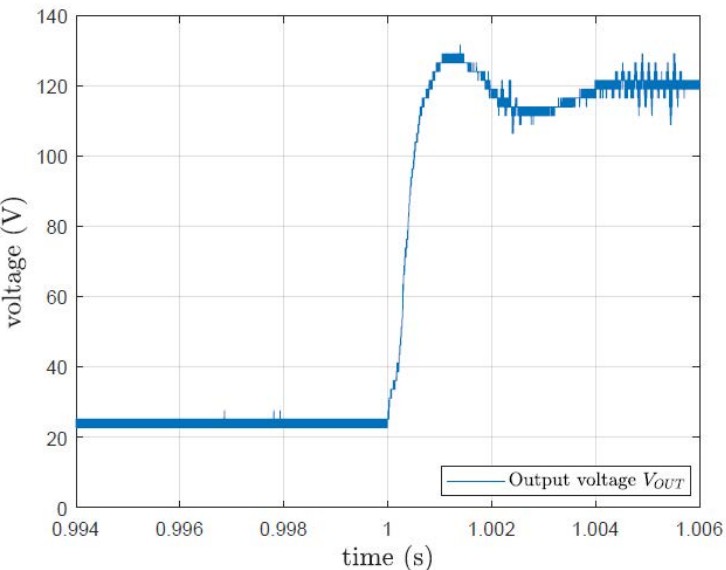

**Figure 16.** $V_{out}$ voltage during a start-up test, with PID controller.

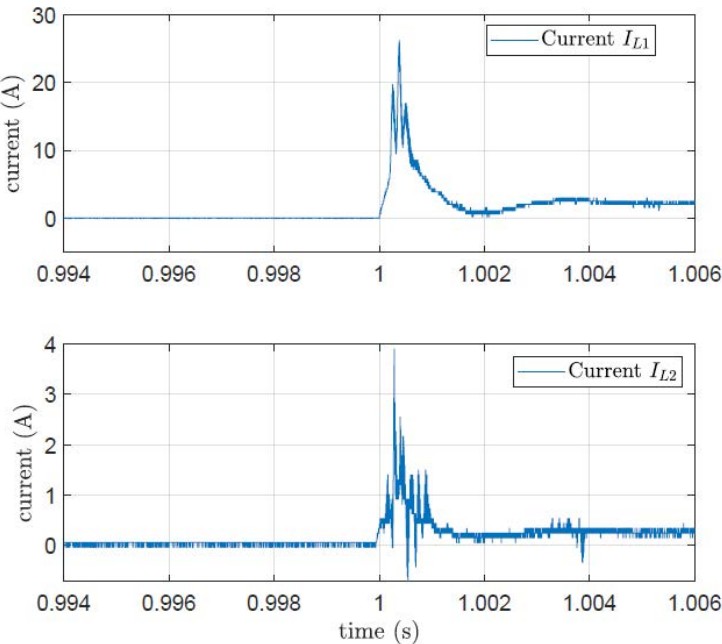

**Figure 17.** $I_{L1}$ and $I_{L2}$ currents during a start-up test, with PID controller.

All these results show the capability of the system to cope with sudden changes in the nominal operating conditions. Indeed, the control system automatically stabilizes, by means of a new sliding regime, the system trajectories of the new equilibrium point. This confirm the effectiveness of the proposed control strategy.

## 6. Conclusions

The paper describes a method for designing a robust control system aimed at regulating the output voltage of a single switch quadratic boost DC–DC converter, despite the possible parameter uncertainties and input voltage and load variations. The inner loop sliding mode controller is very efficient at covering a large range of functions, giving excellent dynamical performance and good steady-state behaviour, without the need of on-line controller parameter tuning. Note that the sliding surface presented in this paper is suitably developed for the considered QBC and is not presented in other works in the literature. The idea of gaining robustness by means of an outer control loop, which modifies

the reference equilibrium state, revealed its efficiency from the experimental point of view. Finally, the implementation of the proposed controller on low-cost hardware shows the opportunity for an immediate transfer from the technological point of view.

**Author Contributions:** Conceptualization, F.A., F.D., G.G. and A.S.; methodology, F.A., F.D., A.F., G.G. and A.S.; software, F.D., G.G. and A.S.; validation, F.A., A.B., M.C., F.D., A.F., G.G., A.A.M., A.S. and S.S.; formal analysis, F.A., F.D., A.F., G.G. and A.S.; investigation, F.A., A.B., M.C., F.D., A.F., G.G., A.A.M., A.S. and S.S.; resources, F.A., A.B., M.C., F.D., A.F., G.G., A.A.M., A.S. and S.S.; data curation, F.A., A.B., M.C., F.D., A.F., G.G., A.A.M., A.S. and S.S.; writing—original draft preparation, F.A., A.B., M.C., F.D., A.F., G.G., A.A.M., A.S. and S.S.; writing—review and editing, F.A., A.B., M.C., F.D., A.F., G.G., A.A.M., A.S. and S.S.; visualization, F.A., A.B., M.C., F.D., A.F., G.G., A.A.M., A.S. and S.S.; supervision, A.S.; project administration, A.B. and A.A.M.; funding acquisition, A.B. and A.A.M. All authors have read and agreed to the published version of the manuscript.

**Funding:** This work was realized into the project frame of REACTION "first and euRopEAn siC eighT Inches pilOt liNe", co-funded by the ECSEL Joint Undertaking under grant agreement No 783158 [25].

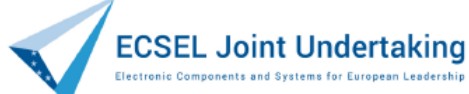 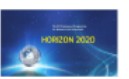 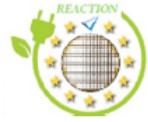

**Conflicts of Interest:** The authors declare no conflict of interest.

**Appendix A**

$$k_{110} = \frac{1}{L_1}; \quad k_{210} = \frac{r_{L1}}{L_1^2}; \quad k_{231} = \frac{1}{(C_1 L_1)}; \quad k_{310} = \frac{r_{L1}^2}{L_1^3}; \quad k_{312} = \frac{1}{(C_1 L_1^2 R_0)}; \quad k_{321} = \frac{1}{(C_1 L_1 L_2)};$$

$$k_{331} = \frac{r_{L1}}{(C_1 L_1^2)}; \quad k_{410} = \frac{C_1 r_{L1}^3}{(C_1 L_1^4)}; \quad k_{412} = \frac{2 L_1 r_{L1}}{(C_1 L_1^4 R_0)}; \quad k_{421} = \frac{r_{L1}}{(C_1 L_1^2 L_2)};$$

$$k_{431} = \frac{(L_1^2 - C_1 L_2 r_{L_1}^2)}{(C_1^2 L_1^3 L_2)}; \quad k_{433} = \frac{L_2 L_1}{(C_1^2 L_1^3 L_2 R_0)}; \quad k_{442} = \frac{1}{(C_1 C_2 L_1 L_2 R_0)};$$

$$k_{211Q} = \frac{k_{210} k_{331} - k_{231} k_{310}}{k_{110} k_{231} k_{321}}; \quad k_{212Q} = \frac{k_{231} k_{312}}{k_{110} k_{231} k_{321}}; \quad k_{221Q} = \frac{k_{331}}{k_{231} k_{321}}; \quad k_{231Q} = \frac{1}{k_{321}};$$

$$k_{311Q} = \frac{k_{210}}{k_{110} k_{231}}; \quad k_{411Q} = \frac{k_{210}(k_{321} k_{431} + k_{331} k_{421})}{k_{110} k_{231} k_{321} k_{442}} - \frac{k_{310} k_{421} + k_{c321} k_{410}}{k_{110} k_{321} k_{442}};$$

$$k_{412Q} = \frac{k_{210} k_{321} k_{433} + k_{231} k_{312} k_{421} - k_{231} k_{321} k_{412}}{k_{110} k_{231} k_{321} k_{442}}; \quad k_{421Q} = \frac{k_{321} k_{431} + k_{331} k_{421}}{k_{231} k_{321} k_{442}};$$

$$k_{422Q} = \frac{k_{321} k_{433}}{k_{231} k_{321} k_{442}}; \quad k_{431Q} = \frac{k_{421}}{k_{321} k_{442}};$$

$$k_{14T} = \frac{k_0}{k_{110}}; \quad k_{241T} = k_{221Q} k_{01} + k_{231Q} k_{20}; \quad k_{242T} = k_{221Q} k_{02} + k_{231Q} k_{21}; \quad k_{243T} = k_0 k_{211Q};$$

$$k_{244T} = k_0 k_{212Q}; \quad k_{341T} = \frac{k_{01}}{k_{231}}; \quad k_{342T} = \frac{k_{02}}{k_{231}}; \quad k_{343T} = k_0 k_{311Q};$$

$$k_{441T} = k_{01} k_{422Q} + k_{431Q} k_{21} + k_{02} k_{421Q}; \quad k_{442T} = \frac{k_{01} k_{421Q} + k_{20} k_{431Q} + k_{30}}{k_{442}};$$

$$k_{443T} = k_{02} k_{422Q}; \quad k_{444T} = k_0 k_{412Q}; \quad k_{445T} = k_0 k_{411Q};$$

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
