# Peer review of "Nonlinear Robust Control of a Quadratic Boost Converter in a Wide Operation Range, Based on Extended Linearization Method"

_electronics, doi:10.3390/electronics11152336_

Round 1
Reviewer 1 Report
1. Why did the authors develop the complicated sliding function of (22) since the controllable canonical form-like model of (13) was constructed?
2. From (4) and (5), it is difficult to derive (11).
3. From (37) it obviously shows that lambda_e is not constant. This causes that (11) should be modified with variable lambda_e.
4. In (22), what were the values of zi and zei taken in the experiments? Was zi fixed in the whole time-response?
5. In (28)-(30), is there a log function? why?
6. There are large overshoot responses in Figs. 8 and 11. Are they due to the complex sliding function?
7. Due to (9), the switching frequency is not constant. How to implement it under a digital controller?
Author Response
1) Why did the authors develop the complicated sliding function of (22) since the controllable canonical form-like model of (13) was constructed?
We added a new Remark 1 (before paragraph 3.4) in order to better justify the choose of the sliding surface and to show that the controllable canonical form model allows to highlight the structure of the sliding hyperplane.
2) From (4) and (5), it is difficult to derive (11).
Thanks for your comments, we include more details for obtaining (11) (see the new part before Eq. (11)).
3) From (37) it obviously shows that lambda_e is not constant. This causes that (11) should be modified with variable lambda_e.
Looking at (7) it appears that $\lambda_e$ can be obtained from the fourth component of the equilibrium state $z^*_e$, i.e. the reference output voltage. For this reason, it is convenient to compute $\lambda_e$ from the desired output voltage, and to compute the remaining components of the equilibrium state (7) for this value $\lambda_e$. In this way, the value of $\lambda _e$ is constant once the equilibrium point is chosen, and it changes only if the reference value of the output voltage has to be changed. The structure of Equation (11) remains the same, but it represents the dynamics of the state referred to the new equilibrium state. We clarify this aspect in the new version of the paper.
4) In (22), what were the values of ziand zei taken in the experiments? Was zi fixed in the whole time-response?
In (22) z_{ei} is the i-th component of the desired equilibrium state and it is a priori known, whereas z_i is the i-th component of the state vector of the converter model, and it is a measured quantity in the sampling instants. This is now specified after Eq (22).
5) In (28)-(30), is there a log function? why?
Expressions (28)-(30) result from the integration process given in (22).
6) There are large overshoot responses in Figs. 8 and 11. Are they due to the complex sliding function?
These overshoots are due to the fact that output capacitor is discharged at the beginning of the experiments. For this reason, in order to obtain a fast output voltage response a high current is required. In any case the current is limited and well tolerated by the converter. The inrush current can be attenuated pre-charging the capacitor. This comment was introduced in the new version of the manuscript.
7) Due to (9), the switching frequency is not constant. How to implement it under a digital controller?
Thanks for your comment, we included a new part in the experimental set-up section and in the introduction, explaining how the controller is implemented. Basically, it has been exploited the peculiar architecture of the TMS320F28379D, that embeds both a CPU and a dedicated control law accelerator. More details are given in the new part of the paper.

Reviewer 2 Report
This paper proposes the SMC and integral type control for the QBC. The comments for the manuscript are as follows:
Please clarify that (37) is using measured Vout or reference Vout? Please unify with the Figs 3 and 4.
Please improve the resolution of the Figs 3 and 4.
Is the lambda in (7) calculated by (37)? Under what conditions of (37), the value of Lambda will be 0 or 1. A simulation result will be interesting for those 2 cases.
Please summarize the experimental results in the tabular form for easier comparison.
The authors should add the comparative results with the conventional techniques.
The introduction lacks the critical review of the previous work. The authors only reported the previous work, but they should also critically review it in introduction section.
Author Response
This paper proposes the SMC and integral type control for the QBC. The comments for the manuscript are as follows:
1) Please clarify that (37) is using measured Vout or reference Vout?
Equation (37) gives the value of $\lambda_e$ corresponding to the reference output $v^{ref}_{out}$, as shown in Fig. 3. This has been clarified after Eq. (37).
2) Please unify with the Figs 3 and 4.
Fig.s 3 and 4 have been unified in a single figure 3.
3) Please improve the resolution of the Figs 3 and 4.
The resolution of the Figs 3 and 4 (now only Fig. 3) has been improved, by replacing the .png figure with a .pdf figure.
4) Is the lambda in (7) calculated by (37)? Under what conditions of (37), the value of Lambda will be 0 or 1. A simulation result will be interesting for those 2 cases.
(7) gives the equilibrium state for a generic value of $\lambda$. (37) gives the value of lambda for a desired output voltage. If we substitute (37) in (7) we obtain the equilibrium state whose fourth component coincides with the desired output voltage. This has been clarified in the new version of the manuscript.
However, note that the value of lambda is never 0 or 1 at steady state, because $\lambda=0$ means that the switch is always turned off and consequently the output voltage is equal to the input voltage. Otherwise, $\lambda=1$ means that the switch is always turned on, the output voltage is zero and the input current increase up to thousands of Amperes since $i_L=Vin/R_L$, and $R_L$ typically is about 10m\Ohm.
5) Please summarize the experimental results in the tabular form for easier comparison.
In the new version of the manuscript Table 2 has been included, by summarizing the main performance indexes for all experimental tests.
6) The authors should add the comparative results with the conventional techniques.
We included the comparison with the conventional PI controller (See the new figures 16 and 17. Moreover these results were adequately commented in the Experimental Results Section.
7) The introduction lacks the critical review of the previous work. The authors only reported the previous work, but they should also critically review it in introduction section.
The introduction has been reviewed and widely rewritten to the best of the Authors capabilities in order to better describe of the previous works with a more critical point of view.

Reviewer 3 Report
Author propose a control system for a quadratic boost DC/DC converter in a wide range of the operations, based on an inner loop with a sliding mode controller, for reaching a desired equilibrium state, and an outer loop with integral-type controller, for assuring robustness against load and input voltage variations and converter parameter uncertainties. It is shown that the proposed controller also deals with the sudden changes in the nominal operating conditions. Thus, if a change of the operating conditions takes place, the proposed control scheme automatically creates a sliding regime which stabilizes the converter trajectories to the new equilibrium point. Experimental results carried out on a suitably developed test set up, show the effectiveness of the proposed approach.
The paper is well-structured and written. The topic is surely interesting and the proposed controller is described in each its particular, and design strategy is well-defined. I have some dubt about the degree of novelty, being the used topology known.
I have only some suggestion to the authors:
1) Abstract contains more than one similar sentence, I think that this part could be slightly abbreviated by compacting some sentence.
2) Referrencing looks not uniform along the paper, sometime it is used ([$$]) and other only [$$]. I suggest to make referrencing uniform for the entire paper.
3) For completeness reason, it should be usefull to define each single parameter of the introduced models, like circuit parameters rL1, rL2. Are these the ESR of the inductors ? If yes, please specify it.
4) Eta in (10) lacks of definition.
5) What's about efficiency of the system ?
Author Response
Author propose a control system for a quadratic boost DC/DC converter in a wide range of the operations, based on an inner loop with a sliding mode controller, for reaching a desired equilibrium state, and an outer loop with integral-type controller, for assuring robustness against load and input voltage variations and converter parameter uncertainties. It is shown that the proposed controller also deals with the sudden changes in the nominal operating conditions. Thus, if a change of the operating conditions takes place, the proposed control scheme automatically creates a sliding regime which stabilizes the converter trajectories to the new equilibrium point. Experimental results carried out on a suitably developed test set up, show the effectiveness of the proposed approach.
The paper is well-structured and written. The topic is surely interesting and the proposed controller is described in each its particular, and design strategy is well-defined. I have some doubt about the degree of novelty, being the used topology known.
Thank you for four evaluation, we tried to better highlight, in this version of the manuscript the main novelties of this work. Basically, the sliding surface presented in this paper is suitably developed for the considered QBC and is not presented in other works in literature to the best of the Author's knowledge. Indeed, the extended linearization procedure was only applied to simpler converter topologies, such as the boost converter, which results in easier sliding surfaces, and tested only in simulation. Whereas, in this work, experimental results will be shown.
I have only some suggestion to the authors:
1) Abstract contains more than one similar sentence, I think that this part could be slightly abbreviated by compacting some sentence.
The abstract has been reviewed and abbreviated in this new version of the manuscript.
2) Referrencing looks not uniform along the paper, sometime it is used ([$$]) and other only [$$]. I suggest to make referrencing uniform for the entire paper.
All the references have been checked and a common format has been used.
3) For completeness reason, it should be usefull to define each single parameter of the introduced models, like circuit parameters rL1, rL2. Are these the ESR of the inductors? If yes, please specify it.
Thank you for this comment, we added a column in table I, specifying the meaning of each symbol.
4) Eta in (10) lacks of definition.
Thanks, it has been defined.
5) What's about efficiency of the system?
In order to evaluate the efficiency of the system, Fig. 9 depicts the measured system efficiency for different values of input power at steady state.

Round 2
Reviewer 1 Report
The last component A1xu of (2) is not present in (4). This means that from (4) and (5) of view, Vin represents the control input of the quadratic boost converter. However, the real control effort is of u in (2). If u is the real control effort for the converter, the matrix A in (4) should contain the term of bVin of (2), isn't it?
Author Response
Reviewer 1.
The last component A1xu of (2) is not present in (4). This means that from (4) and (5) of view, Vin represents the control input of the quadratic boost converter. However, the real control effort is of u in (2). If u is the real control effort for the converter, the matrix A in (4) should contain the term of bVin of (2), isn't it?
Thanks for this comment. The Reviewer is right saying that u is the real control effort for the converter. However, when the averaged model (4)-(5) is obtained (using the procedure illustrated in [8] and [20]) the new real input is $\lambda$ and model (4)-(5) coincides with model (2)-(3) when $\lambda=u=0$ (OFF state) and when $\lambda=u=1$ (ON state). The variable $\lambda$ is embedded in the definition of matrix A, as shown after eq. (5). Actually, this aspect was not sufficiently clear as pointed out by this reviewer. For this reason, we include the explicit dependence of matrix A from $\lambda$, and we also clarify the correspondence between model (2)-(3) and (4)-(5) in the current version of the manuscript.
Reviewer 2 Report
The authors have significantly improved the manuscript. Although the authors compared the results with conventional PID in Figs. 16 and 17. However, the comparative analysis of the results with the Figs. 6 and 7 is missing. Also, the comparative analysis of the numbers should be added in Table 2 for fair comparison.
Author Response
Reviewer 2.
The authors have significantly improved the manuscript. Although the authors compared the results with conventional PID in Figs. 16 and 17. However, the comparative analysis of the results with the Figs. 6 and 7 is missing. Also, the comparative analysis of the numbers should be added in Table 2 for fair comparison.
Thank for this comment. The results have been adequately analyzed and commented (from the comparative point of view) in the current version of the manuscript. Moreover, the results with conventional PI have been included in Table 2 as well, in order to be easily compared with the other results.
Round 3
Reviewer 2 Report
The authors have reflected my comments. Thanks.